# Attention Measurement of an Autism Spectrum Disorder User Using EEG Signals: A Case Study

José Jaime Esqueda-Elizondo [1,2], Reyes Juárez-Ramírez [2], Oscar Roberto López-Bonilla [1], Enrique Efrén García-Guerrero [1], Gilberto Manuel Galindo-Aldana [3], Laura Jiménez-Beristáin [2], Alejandra Serrano-Trujillo [2], Esteban Tlelo-Cuautle [4] and Everardo Inzunza-González [1,*]

1 Facultad de Ingeniería, Arquitectura y Diseño, Universidad Autónoma de Baja California, Carretera Transpeninsular Ensenada-Tijuana No. 3917, Ensenada C.P. 22860, Baja California, Mexico; jjesqueda@uabc.edu.mx (J.J.E.-E.); olopez@uabc.edu.mx (O.R.L.-B.); eegarcia@uabc.edu.mx (E.E.G.-G.)

2 Facultad de Ciencias Químicas e Ingeniería, Universidad Autónoma de Baja California, Calzada Universidad No. 14418, Parque Industrial Internacional, Tijuana C.P. 22390, Baja California, Mexico; reyesjua@uabc.edu.mx (R.J.-R.); ljimenezb@uabc.edu.mx (L.J.-B.); aserrano11@uabc.edu.mx (A.S.-T.)

3 Facultad de Ciencias de la Ingeniería y Tecnología, Universidad Autónoma de Baja California, Carretera Estatal No. 3, Gutiérrez, Mexicali C.P. 21720, Baja California, Mexico; gilberto.galindo.aldana@uabc.edu.mx

4 Departamento de Electrónica, Instituto Nacional de Astrofísica, Óptica y Electrónica, Luis Enrique Erro No. 1, Santa María Tonanzintla, Puebla C.P. 72840, San Andrés Cholula, Mexico; etlelo@inaoep.mx

* Correspondence: einzunza@uabc.edu.mx; Tel.: +52-646-175-0744

**Abstract:** Autism Spectrum Disorder (ASD) is a neurodevelopmental life condition characterized by problems with social interaction, low verbal and non-verbal communication skills, and repetitive and restricted behavior. People with ASD usually have variable attention levels because they have hypersensitivity and large amounts of environmental information are a problem for them. Attention is a process that occurs at the cognitive level and allows us to orient ourselves towards relevant stimuli, ignoring those that are not, and act accordingly. This paper presents a methodology based on electroencephalographic (EEG) signals for attention measurement in a 13-year-old boy diagnosed with ASD. The EEG signals are acquired with an Epoc+ Brain–Computer Interface (BCI) via the Emotiv Pro platform while developing several learning activities and using Matlab 2019a for signal processing. For this article, we propose to use electrodes F3, F4, P7, and P8. Then, we calculate the band power spectrum density to detect the Theta Relative Power (TRP), Alpha Relative Power (ARP), Beta Relative Power (BRP), Theta–Beta Ratio (TBR), Theta–Alpha Ratio (TAR), and Theta/(Alpha+Beta), which are features related to attention detection and neurofeedback. We train and evaluate several machine learning (ML) models with these features. In this study, the multi-layer perceptron neural network model (MLP-NN) has the best performance, with an AUC of 0.9299, Cohen's Kappa coefficient of 0.8597, Matthews correlation coefficient of 0.8602, and Hamming loss of 0.0701. These findings make it possible to develop better learning scenarios according to the person's needs with ASD. Moreover, it makes it possible to obtain quantifiable information on their progress to reinforce the perception of the teacher or therapist.

**Keywords:** autism; attention; ASD; learning activities; EEG; BCI; features; artificial intelligence; machine learning

## 1. Introduction

Scientists have always been captivated by the brain, and cognitive processes are also the most intriguing for most people. A fundamental part of these cognitive processes is the attention process. To obtain knowledge, first, the attention process is needed. Attention is a cognitive process that enables selecting, focusing on, and sustained information processing [1]. The object of attention can either be an environmental stimulus actively processed

by sensory systems or associative information and response alternatives generated by the ongoing cognitive activity. This allows us to orient ourselves towards relevant stimuli, ignoring those not, and act accordingly. Moreover, it is the basis of learning, and it is necessary to have it, in order to begin the learning process. There have been many measuring techniques, such as using the response times or the number of clicks given while using particular software, the eye contact time measured from videos, Magnetic Resonance Imaging (MRI) or functional Magnetic Resonance Imaging (fMRI) studies, among other techniques. Autism Spectrum Disorder (ASD) is a neurodevelopmental life condition characterized by problems with social interaction, low verbal and non-verbal communication skills, and repetitive and restricted behavior [2,3]. People with ASD usually have variable attention levels because they have hypersensitivity and large amounts of environmental information are a problem for them.

There are many methods for measuring attention reported in the literature, such as eye-tracking/gaze [4,5], fMRI [6,7], using a program [8], biofeedback [9], and electroencephalographic (EEG) signals [10–12], among others. The last one delivers great advantages over other neuroimaging techniques due to its high temporal resolution [13], neurodevelopmental diagnosis accuracy [14], cognitive-related bioelectrical data [15], low cost [16], and non-invasive application methods [17]. The authors [8] show how the attention of 49 children with ASD and a group of 51 typical children is measured using a mindfulness-based program (MBP); in other words, this is a computerized attention test. This MBP software measures the accuracy and reaction times, but they did not directly measure. Another way to measure attention is by analyzing the facial expressions or measuring the timing of eye contact from video recordings. The study [5] shows a measuring technique based on the analysis of video recordings of 1756 toddlers from 12 to 72 months with ASD while watching selected short videos on an iPhone or an iPad. Their facial expressions are video-recorded and analyzed as they watch the videos. Reference [18] presents a study about the concentration measurement of a children group while interacting with an NAO robot and their teacher. In this case, the eye contact time was measured by analyzing the video recordings of the sessions obtained with two cameras at the posterior.

The study [19] shows an approach to the joint analysis of EEG and eye-tracking for children's ASD evaluation. First, the synchronization measures, information entropy, and time-frequency features of the multi-channel EEG are derived. Then, a random forest is applied to the eye-tracking recordings of the same subjects to single out the most significant features. A convolutional graph network (GCN) model naturally fuses the two groups of features to differentiate the children with ASD from the typically developed (TD) subjects. Reference [20] uses EEG activity (raw EEG and alpha power) to provide a time-resolved index of attentional orienting towards salient stimuli that either matched or did not match target-defining properties. In all of the references presented above, the use of feature extraction techniques helps to obtain information from the signals acquired. These feature extraction techniques can help us to obtain useful or descriptive information while eliminating or reducing redundant or unnecessary information, noise, or artifacts. Once the feature extraction stage has finished, the classification can quantify the signals. This paper also shows the feature extraction and the classification algorithms most frequently used.

Nowadays, intelligent systems that incorporate artificial intelligence (AI) frequently rely on machine learning (ML) [21,22]. ML is a term that refers to a system's ability to learn from problem-specific training data in order to automate the process of developing analytical models and completing associated tasks [23,24]. Deep learning (DL) is a paradigm in machine learning that is based on the use of artificial neural networks [25,26]. Commonly, the use of ML algorithms is centered in the diagnosis or detection of ASD, as is presented in [20]. The authors in [27] used EEG and eye-tracking features to identify children with ASD. In [28], the authors used deep convolutional architectures to detect ASD. Other studies [29] reported statistical features for ASD classification. In reference [30], they used an ML and a DL process for diagnosing ASD from time-frequency spectrogram images of EEG. The authors in [31] reported that it is possible to evaluate mental stress using DL and

EEG records. There are also studies such as [32], where they used the free artifact signal of two electrodes to detect ASD. In [33], they used a hybrid light-weighted feature extractor from signal to spectrogram images.

Recent studies have a focus on the relationship between human and machine behavior, based on the premise that diverse social and psychological backgrounds correspond in practice with different modalities of human–computer interaction [34]. In general, EEG feature extraction techniques have offered strong clinical consistency since the beginning of their use for assessing and diagnosing different cognitive and neurological domains in ASD [35], learning difficulties [36], and attention [37]. It is widely accepted that AI techniques are helpful for automatic diagnosis and rehabilitation procedures in ASD cases. For example, in [38], a review of DL methods focusing on neuroimaging-based approaches is presented. Furthermore, the authors report a review of studies based on DL networks for diagnosing ASD and the challenges in automatized detection and ASD rehabilitation. Nowadays, there are some DL applications for brain disease diagnoses, such as the ones presented in [39] , which presents a review of automated multiple sclerosis (MS) detection methods based on MRI. They notice that the most used architectures for MS detection are convolutional neural networks (CNNs), autoencoders (AEs), generative adversarial networks (GANs), and CNN-RNN models. Schizophrenia (Sz) is another brain disease detected with DL methods using EEG signal processing [40]. The authors compare their results with the traditional AI methods, such as support vector machine (SVM), k-nearest neighbors, decision tree, naïve Bayes, random forest, extremely randomized trees, and bagging. The DL models used are long short-term memories (LSTMs), one-dimensional convolutional networks (1D-CNNs), and 1D-CNN-LSTMs. Convolutional neural networks and LSTMs perform best, cross-validated with a k-fold of 5. Moreover, epileptic seizures are detectable by using EEG signal processing; for example, in [41], the authors present a novel diagnostic procedure that uses fuzzy theory and DL techniques. They propose an adaptive neuro-fuzzy inference system (ANFIS) with a breeding swarm optimization (BS) method. These ANFIS-BS methods present accuracy of 99.74 % in a two-class classification task. Appendix A summarizes in Tables A1 and A2 the state of the art and shows a comparison with the proposed method, considering the dataset, data source, preprocessing, methods/algorithm, main findings, and applications.

The research questions that motivate this paper are: (1) What brain regions activate on average when attention increases? At what levels? Depending on the type of activity to be developed? (2) Can the level of the attention span of a person with Autism Spectrum Disorder be quantified as a feature using time-frequency analysis methods? (3) Is there a relationship between the increment in the power of electroencephalographic signals and attention span in a child with Autism Spectrum Disorder?

In this paper, the hypothesis is that measuring and quantifying the brain's electrical activity (power spectrum density) makes it possible to assess the level of attention when performing various cognitive activities and interacting with different software or systems. Therefore, this article aims to detect when an ASD user has high attention levels while developing learning activities based on the EEG signals acquired by an Epoc+ Brain–Computer Interface (BCI). The novelty of this paper is the use of ML algorithms to classify the "Attention" and "No Attention" states of an ASD user. This research presents a new methodology based on EEG signals and ML algorithms for classifying the attention of a 13-year-old boy with ASD. This research formulates a method for processing electroencephalographic signals to determine attention lapses in people with ASD, tested by performing various learning activities and interacting with computer programs.

The rest of this paper is organized as follows. Section 2 presents the materials and the proposed methodology. Section 3 shows the findings of this paper. Section 4 presents the discussion. Finally, Section 5 summarizes our conclusions.

## 2. Materials and Methods

The approval of this research by the Ethics Committee and Research for Pre-Graduates and Post-Graduates of the Facultad de Ingeniería y Negocios Guadalupe Victoria de la Universidad Autónoma de Baja California was obtained on 8 October 2020, with the POSG/020-1-04 register. The EEG signals were acquired with an Epoc+ Brain–Computer Interface (BCI) [42,43] via the Emotiv Pro platform while the ASD user developed several learning activities, and data were processed with Matlab 2019a and Emotiv Pro software using the Student Version.

Figure 1 depicts the electrode location (left) and the Emotiv Epoc+ headset (right). According to the coherence analysis in attention [44,45], the selected electrodes were F3, F4, P7, and P8.

The proposed methodology and the simulations were performed on a personal computer with the following specifications: Intel(R) Core i5-8250U CPU @ 1.60 GHz, 1800 Mhz, 4 Cores, 8 Logical Processors, and 8 GB in RAM.

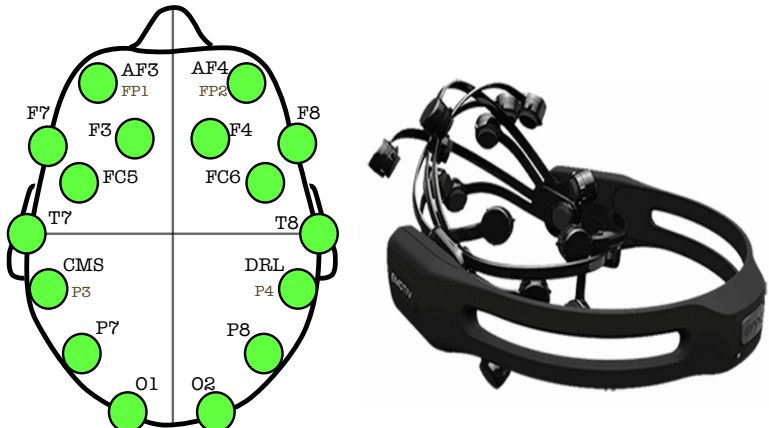

**Figure 1.** Electrode location (**left side**) of the Epoc+ headset (**right side**) of the Emotiv Inc., taken from Emotiv website https://emotiv.gitbook.io/epoc-user-manual/, accessed on 29 December 2021.

The signal was sampled at 2048 Hz, filtered with a dual-notch filter at 50 Hz and 60 Hz and a low-pass filter at 64 Hz, and then downsampled to 128 Hz for transmission. It was necessary to multiply the signal by 0.51 μ to convert it to a voltage.

The proposed data acquisition process is as follows:

Step 1.  Place the headset with the electrodes hydrated on the test subject.
Step 2.  Start the video recording and the EEG data acquisition.
Step 3.  Give the worksheet to the test subject and the instructions.
Step 4.  Let the test subject start the activity, and give him additional instructions if necessary, as in a regular school session.
Step 5.  When the activity is over, stop video recording and data acquisition.

Figure 2 shows the EEG acquisition process and how the boy worked with the activity sheets using the Epoc+ headset.

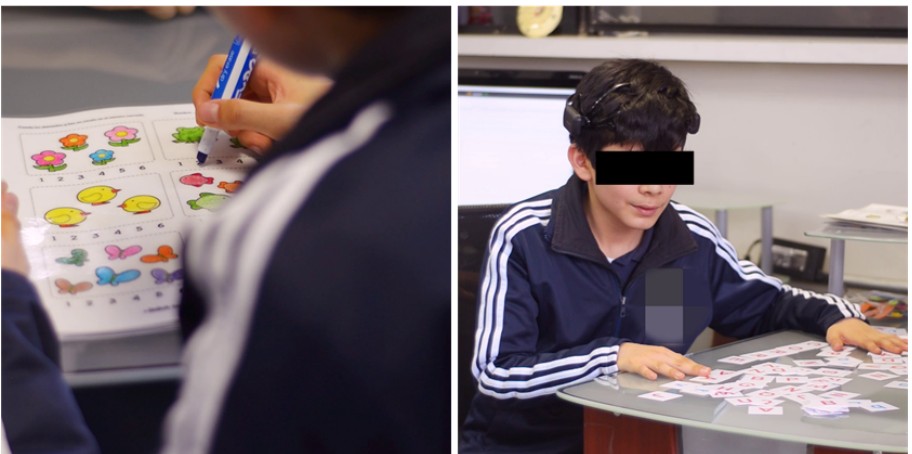

**Figure 2.** Data acquisition process with the Emotiv Epoc+. The EEG recordings start once the localization of the headset is correct, and the signal quality, and the electrode contacts are verified and in green level. Pictures are from http://imagentv.uabc.mx/videos/electro-encefalograf%C3%ADas-y-autismo-uabc-no-se-detiene-imago, accessed on 29 December 2021.

### 2.1. Activity Sheets

Figures 3 and 4 depict examples of other activity sheets provided by the child's teachers, according to his knowledge and abilities. Figure 3 shows an activity sheet about reading, following instructions, and drawing. Figure 4 is a counting animal activity sheet. The school for children with ASD Eduke (https://www.facebook.com/EDUKE-123602824381330, accessed on 29 December 2021), located in Tijuana, Baja California, México, provided all the activity sheets used in this research.

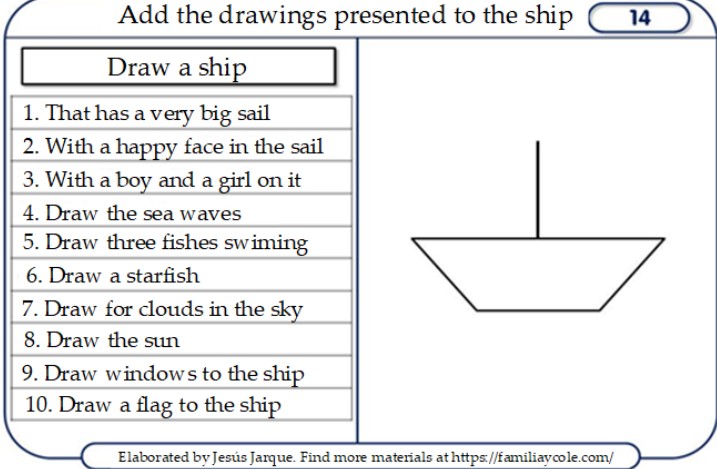

**Figure 3.** Example of reading, following instructions, and drawing activity sheet. This activity requires the child to read and follow instructions. The activity sheets are from https://familiaycole.com/, accessed on 29 December 2021.

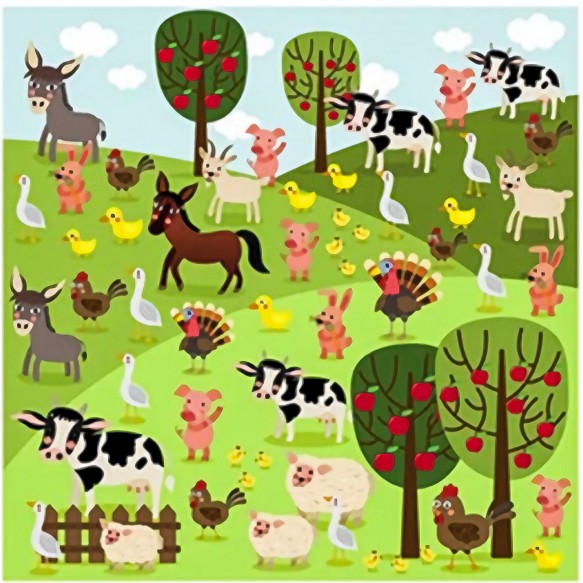

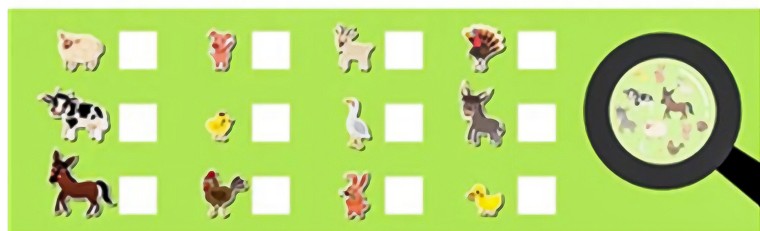

**Figure 4.** Example of counting animals activity sheet. This activity requires the child to identify, classify, count the animals, and write the number in the white square. The activity sheets are from https://www.actividadesdeinfantilyprimaria.com/, accessed on 29 December 2021.

## 2.2. Signal Processing Procedure

Figure 5 depicts the block diagram of the procedure used for signal processing. The first step is preprocessing the EEG signal, and then the power spectrum density of signals is calculated and separated into bands. Next, we obtain the features presented in Table 1 and validate them. With these features, we train the machine learning algorithms. In the next section, we give more information about these steps.

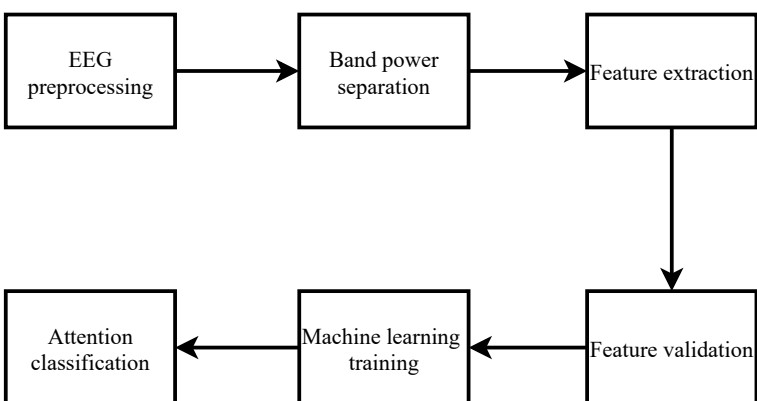

**Figure 5.** Block diagram of the proposed method. The first stage is signal preprocessing, after the band power separation, and then the feature extraction stage. Next is the feature validation process, the machine learning training stage, and finally, the attention quantification result.

### 2.2.1. Preprocessing of EEG Signal

The Emotiv software gives the recordings in a .csv file with integer numbers. It is necessary to convert the EEG signal acquired by the Epoc+ to its voltage equivalent by multiplying it by the factor $0.51 \times 10^{-6}$.

### 2.2.2. Band Power Separation

In EEG signal processing, it is common to separate the power spectrum density into the following bands: Delta (1–4 Hz), Theta (4–8 Hz), Alpha (8–12 Hz), Beta (12–30 Hz), and Ram (or Gamma) (30–50 Hz), depicted in Figure 6. These band powers [46] are the basis for calculating relative powers and ratios in the feature extraction stage. The Emotiv software gives the power of each band, except for the Delta band, and it gives the Beta band separated into Low Beta and High Beta powers [47]. For this research, we add both Beta band powers.

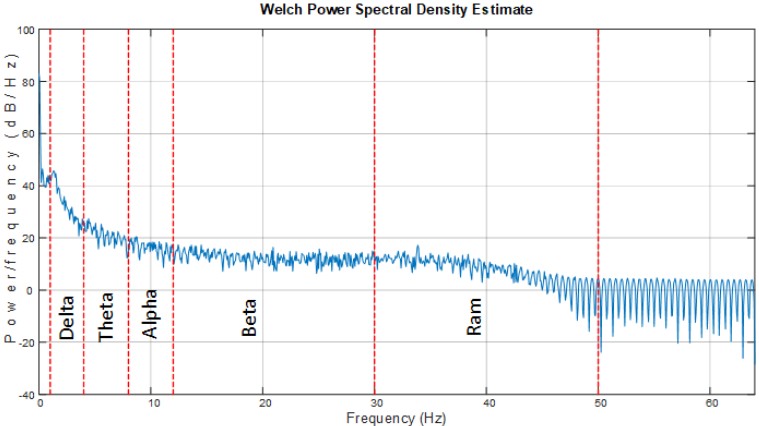

**Figure 6.** Band power separation example, Welch power spectral density estimate (illustrative figure).

The Emotiv software uses two-second windows to calculate the power spectrum density in absolute values, with units $\mu\text{V}^2/\text{Hz}$, and then separates it into bands. The two-second window involves 256 samples [47,48].

Figure 7 shows an example of band power separation. For this paper, we use the electrodes F3, P7, F4, and P8 because they show high coherence in attention tasks [44,45].

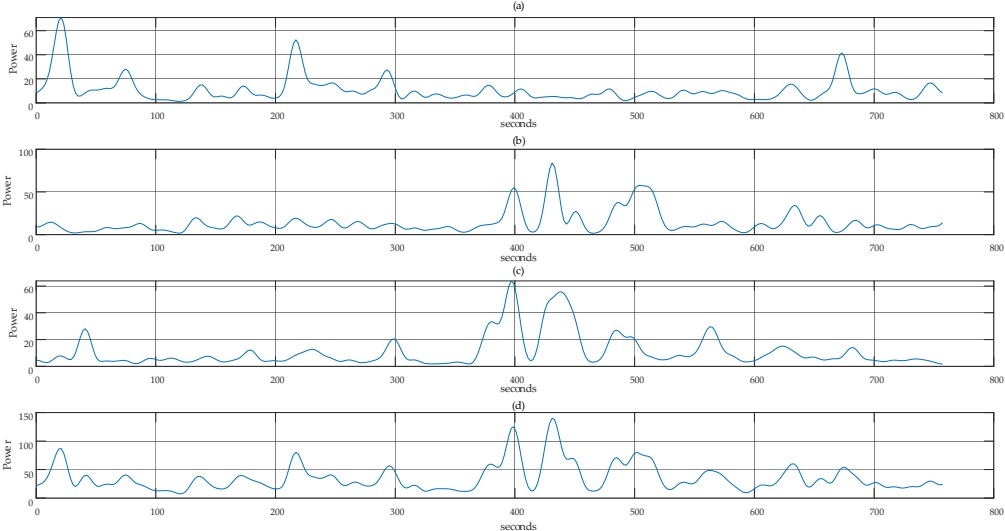

**Figure 7.** Band power separation example from F4 electrode. (**a**) Theta band power, (**b**) Alpha band power, (**c**) Beta band power, (**d**) Total band power.

### 2.2.3. Feature Extraction

To detect the Theta–Beta Ratio (TBR) and the Theta–Alpha Ratio (TAR), it is necessary first to calculate the band power spectrum density (PSD) of the EEG signal in two-second windows and for each channel or electrode. It is common to use the TBR features in attention detection and neurofeedback and the Theta Relative Power Beta and Theta/(Alpha + Beta), known as TBAR [48].

Table 1 presents the features calculated and their equations [48]. The next step is to use these features to train several machine learning models and evaluate their performance.

**Table 1.** Feature equations for attention detection.

| Feature | Equation |
|---|---|
| Theta Relative Power | $TRP = \frac{\theta}{T}$ |
| Alpha Relative Power | $ARP = \frac{\alpha}{T}$ |
| Beta Relative Power | $BRP = \frac{\beta}{T}$ |
| Theta–Beta Ratio | $TBR = \frac{\theta}{\beta}$ |
| Theta–Alpha Ratio | $TAR = \frac{\theta}{\alpha}$ |
| $\frac{Theta}{Alpha + Beta}$ | $TBAR = \frac{\theta}{\beta + \alpha}$ |

$T = \theta + \alpha + \beta$ is the total power [48].

Figure 8 depicts the Theta, Alpha, and Beta relative powers (R.P.) obtained for the F4 electrode using the equations presented in Table 1. These R.P. values change with the time and function of the activity performance. Figure 9 shows the Theta–Beta Ratio, Theta–Alpha Ratio, and Theta/(Alpha–Beta Ratio) for the same F4 electrode.

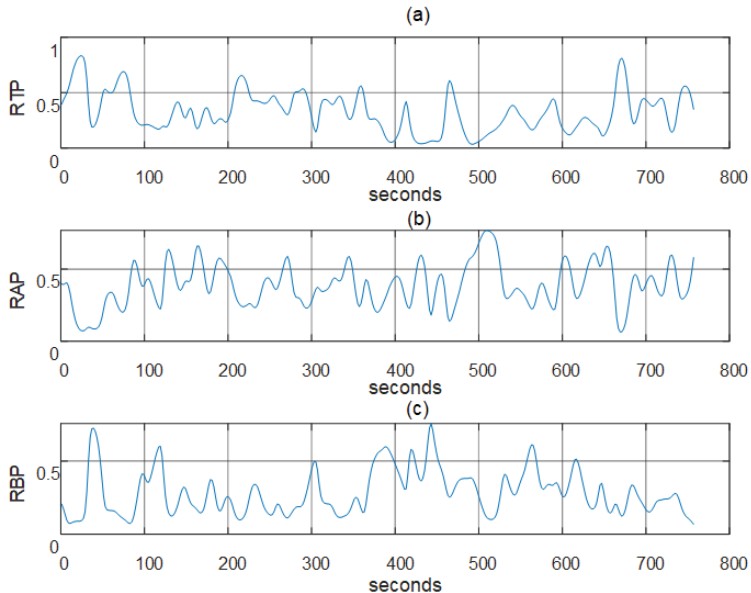

**Figure 8.** Example of relative powers obtained from F4 electrode. (**a**) F4 Theta relative power, (**b**) F4 Alpha relative power, and (**c**) F4 Beta relative power.

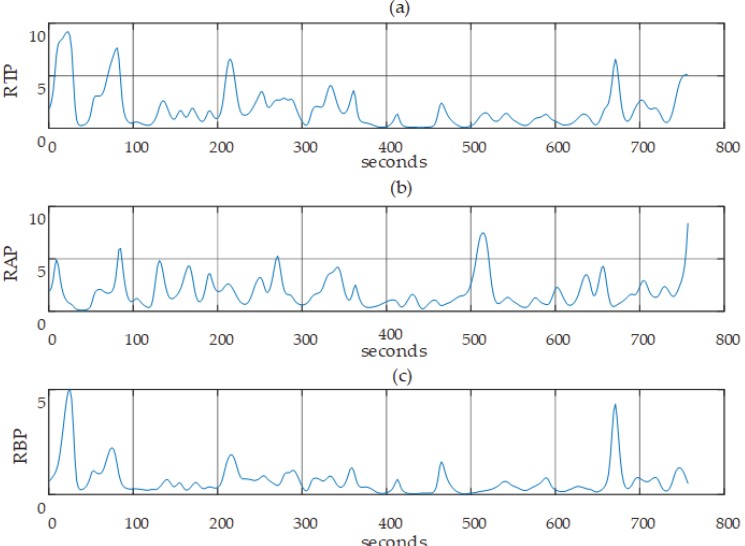

**Figure 9.** Example of ratios obtained from F4 electrode. (**a**) Theta–Beta Ratio, (**b**) Theta–Alpha Ratio, (**c**) Theta/(Alpha–Beta Ratio).

### 2.2.4. Dataset Preparation

The dataset consists of 24 features, 6 features for each electrode, with four electrodes (F3, F4, P7, and P8) and two classes: "Attention" and "No Attention". The dataset has 33,936 samples; it has 16,968 samples for each class to conserve balance. Figure A1 from Appendix B shows a fragment of the created dataset with 24 features acquired through the processing of EEG signals when the user is performing didactic activities and paying attention and when he is not paying attention to his learning process.

The Supplementary Materials dataset included six different Attention activities (counting, forming words, completing words, looking for differences between two figures, reading text, and answering simple questions from the reading), taken in 6 different moments. There are also No Attention samples recorded in non-learning activities such as watching cartoons, echolalia, doing nothing, and just sitting awake, trying to be as relaxed as possible.

### 2.2.5. Machine Learning Algorithm Training

In this paper, we chose eight ML algorithms to evaluate the classification of attention through the EEG signals of an ASD user. The chosen ML algorithms were naive Bayes (N.B.), stochastic gradient descent (SGD), decision trees (D.T.), support vector machine (SVM)-RBF, k-nearest neighbors (KNN), multi-layer perceptron neural network (MLP-NN), random forest (R.F.), and extra trees (E.T.). These ML models are part of the Scikit Learn library [49]. Figure 10 shows the flowchart to perform the training test of the ML algorithms. First, it is necessary to import the libraries or toolboxes required, such as Scikit Learn, Pandas, and Seaborn. Then, the features dataset is loaded; subsequently, separating the input data (features) from the output data is necessary. Next, we randomly divide the dataset, 80% for training and 20% for tests. Then, the data are scaled between 0 and 1 to obtain optimized results. Then, the machine learning model is trained. Then, we perform the scoring of the ML model, i.e., using the confusion matrix and performance metrics to evaluate the ML models.

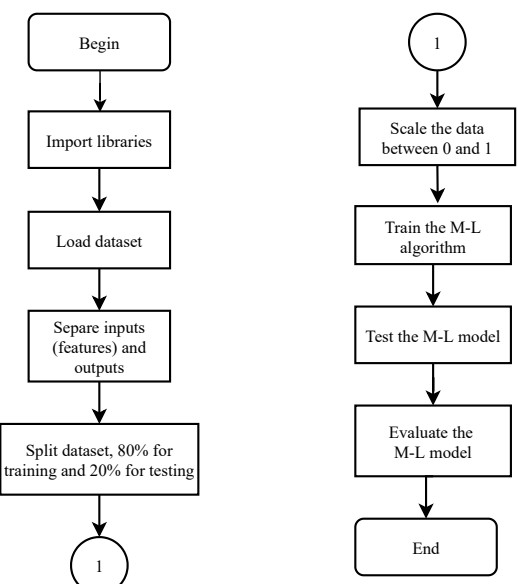

**Figure 10.** Flowchart for training and testing of ML algorithms.

## 3. Results

To evaluate the ML models, we rely on the metrics of the Scikit Learn library [49]. The metrics used to evaluate the scoring of the ML models are the confusion matrix (true positives, true negatives, false positives, false negatives), accuracy, F1 score, precision, sensitivity/recall, and specificity.

Table 2 shows the scoring parameters obtained for the ML models tested in this paper. The first four parameters correspond to the results of the confusion matrix. Naive Bayes with an accuracy of 0.7628, SGD with 0.8619, decision tree with 0.8697, SVM-RBF with 0.8940, KNN with 0.8968, MLP-NN with 0.9298, random forest with 0.9291, and finally extra trees with an accuracy of 0.9270. Therefore, the extra trees model has the best accuracy score.

Regarding the F1 score parameter, it is observable that naive Bayes, SGD, decision trees, and SVM-RBF obtained a score lower than 0.90. Meanwhile, the KNN, MLP-NN, random forest, and extra trees models obtained a score greater than 0.90, with extra trees achieving the highest score. Regarding the specificity/precision, we observed that the naive Bayes model was the lowest, while the extra trees and MLP-NN models were the highest, with 0.8896 and 0.9155, respectively. Regarding the sensitivity/recall score, all the models obtained a result greater than 0.90, except decision trees with 0.8720, and the extra trees model achieved the best result with 0.9738.

Table 3 shows the performance metrics obtained for each ML model. The metrics used to evaluate the performance of the ML models were the Area Under the Curve (AUC), the Cohen's Kappa coefficient, Hamming loss, and the Matthews correlation coefficient. Regarding the AUC metric, we notice that the naive Bayes, stochastic gradient descent, and decision trees models are the lowest, with 0.7642, 0.8624, and 0.8697, while the support vector machine (SVM)-RBF, KNN, extra trees, MLP-NN, and random forest (R.F.) models are the ones that obtained the best AUC, with 0.8944, 0.8972, 0.9274, 0.9299, and 0.9294, respectively, with the MLP-NN model obtaining a better AUC. This measure compares labelings by different human annotators, not a classifier versus ground truth, regarding Cohen's Kappa coefficient. The Kappa score is a number between $-1$ and 1. Scores above 0.8 indicate good agreement; zero or lower means no agreement (practically random labels). We observe that the naive Bayes, stochastic gradient descent, decision trees, support vector machine (SVM)-RBF, and KNN models obtained a Kappa coefficient less than 0.80 but greater than zero. However, the extra trees, MLP-NN, and random forest (R.F.) models obtained Kappa coefficients of 0.8542, 0.8597, and 0.8583, respectively, which are more significant than 0.80. Therefore, it means that these ML models have good

agreement. We notice that the model MLP-NN is the one that obtained the highest Cohen's Kappa coefficient.

**Table 2.** Scoring parameters of the ML algorithms evaluated in this study.

| | Machine-Learning Algorithm | | | | | | | |
|---|---|---|---|---|---|---|---|---|
| **Scoring Parameters** | **Naive Bayes** | **SGD** | **Decision Trees** | **(SVM)-RBF** | **KNN** | **MLP-NN** | **Random Forest (RF)** | **Extra Trees** |
| True positive | 1984 | 2720 | 2967 | 2874 | 2892 | 3126 | 3039 | 3013 |
| True negative | 3194 | 3131 | 2937 | 3195 | 3196 | 3186 | 3268 | 3280 |
| False positive | 1436 | 700 | 453 | 546 | 528 | 294 | 381 | 407 |
| False negative | 174 | 237 | 431 | 173 | 172 | 182 | 100 | 88 |
| Accuracy | 0.7628 | 0.8619 | 0.8697 | 0.8940 | 0.8968 | 0.9298 | 0.9291 | 0.9270 |
| F1 Score | 0.7986 | 0.8698 | 0.8691 | 0.8988 | 0.9012 | 0.9304 | 0.9314 | 0.9278 |
| Specificity/Precision | 0.6898 | 0.8172 | 0.8663 | 0.8540 | 0.8582 | 0.9155 | 0.8955 | 0.8896 |
| Sensitivity/Recall | 0.9483 | 0.9296 | 0.8720 | 0.9486 | 0.9489 | 0.9459 | 0.9703 | 0.9738 |

**Table 3.** Performance metrics of the eight ML algorithms evaluated in this study.

| | Performance Metrics | | | |
|---|---|---|---|---|
| **Machine Learning Algorithm** | **AUC** | **Cohen's Kappa Coefficient** | **Hamming Loss** | **Matthews Correlation Coefficient** |
| Naive Bayes | 0.7642 | 0.5269 | 0.2371 | 0.5674 |
| Stochastic Gradient Descent | 0.8624 | 0.7241 | 0.1380 | 0.7310 |
| Decision Trees | 0.8697 | 0.7395 | 0.1302 | 0.7395 |
| Support Vector Machine (SVM)-RBF | 0.8944 | 0.7883 | 0.1059 | 0.7931 |
| KNN | 0.8972 | 0.7939 | 0.1031 | 0.7983 |
| Extra Trees | 0.9274 | 0.8542 | 0.0729 | 0.8580 |
| MLP-NN | 0.9299 | 0.8597 | 0.0701 | 0.8602 |
| Random Forest (RF) | 0.9294 | 0.8583 | 0.0708 | 0.8613 |

Regarding the Hamming loss, this Hamming loss should be zero; that is, the closer it is to zero, the model tends to be perfect or ideal. In this case, the extra trees, MLP-NN, and random forest (R.F.) models have the lowest Hamming loss. The MLP-NN model has the lowest Hamming loss, with 0.0701. We use in machine learning the Matthews correlation coefficient (MCC) or phi coefficient as a measure of the quality of binary (two-class) classifications, introduced by biochemist Brian W. Matthews [50]. In this case, the three best models are extra trees, MLP-NN, and random forest (R.F.), with 0.8580, 0.8602, and 0.8613, respectively, with random forest being the best (R.F.).

Figure 11 depicts the ROC curve of the top five ML models trained for attention classification using EEG data. The ROC curve shows the trade-off between sensitivity (TPR) and specificity (1-FPR). Classifiers that give curves closer to the top-left corner indicate better performance. The closer the curve comes to the 45-degree diagonal of the ROC space, the less accurate the test is. The SVM-RBF and KNN models are closer to the 45-degree diagonal, resulting in less accuracy. On the other hand, the random forest, extra trees, and MLP-NN models are closest to the upper left. Therefore, they are the ones with the best performance.

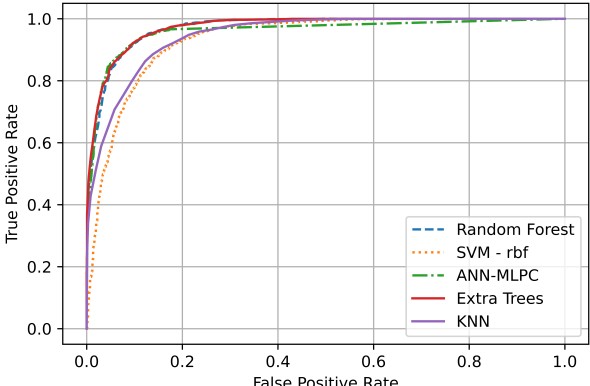

**Figure 11.** The receiver operating characteristic curve (ROC) of the top five ML models trained for attention classification using EEG data.

Figure 12 depicts the training time of the eight ML models tested in this study. The N.B., SGD, KNN, and D.T. models have the shortest training time. However, according to the results shown in Tables 2 and 3, they have the lowest performance metrics. In contrast, the SVM-RBF, R.F., and MLP-NN models have a longer training time of 17.01, 21.14, and 73.10 s, with the MLP-NN model having a longer training time. However, the model also has better performance metrics, as shown in Tables 2 and 3. Therefore, the classifier designer must conduct a cost–benefit analysis in terms of accuracy and processing time. In most cases, programmers prefer better accuracy, sacrificing training time since this process (training) is only done once and only uses the trained model. For this reason, in this study, it would be more convenient to choose the MLP-NN model.

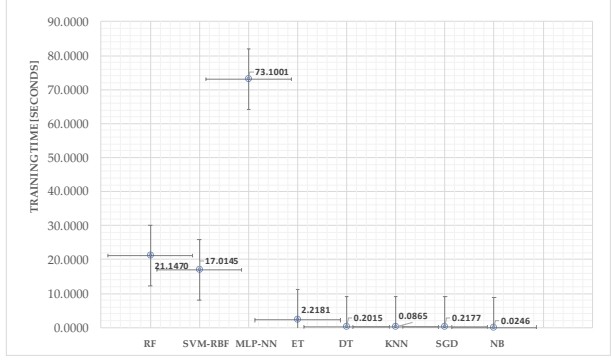

**Figure 12.** Training time of the eight ML models evaluated in this study.

## 4. Discussion

In this research, we observed that the power spectrum density (PSD) is helpful for attention detection, as proposed in the hypothesis. The features based on band PSD, such as Relative Theta Power (RTP), Relative Alpha Power (RAP), Relative Beta Power (RBP), Theta–Beta Ratio (TBR), Theta–Alpha Ratio (TAR), and the TBAR are good features for attention classification. With these features, the multi-layer perceptron neural network model (MLP-NN) achieved the best performance, with an AUC of 0.9299, Cohen's Kappa coefficient of 0.8597, Matthews correlation coefficient of 0.8602, and Hamming loss of 0.0701. Nevertheless, MLP-NN requires a longer training time of up to 73.1 s. However, the results presented in Tables 2 and 3 and Figures 11 and 12 show that the random forest and extra trees models have good performance metrics and a training time of 21.14 and 2.21, respectively. Therefore, the classifier designer must perform a cost–benefit analysis in terms of accuracy and processing time. In most cases, designers prefer better accuracy, sacrificing training time since this process (training) is only performed once, and then only the trained model is used. For this reason, in this study, it would be more convenient to choose the MLP-NN model.

Furthermore, feature extraction improves the acquisition of relevant information for accuracy for diagnosis and has been widely applied to different neuropsychological and neurophysiological fields [51]; the type of waveforms and definition of the morphology of EEG patterns increases the amount of available information for clinical decision making from brain dysfunction [52] to cognitive impairment [53]. Particular interest has been historically directed to the frontal areas in attention measurement, as they correspond to the brain regions responsible for activity direction and orientation. Classification of features may help to describe cortical connectivity, particularly for attentional deficits associated with frontal theta in children [36]. Other research refers to frontal bilateral theta waves in resting EEG in children with learning difficulties and an association with bilateral synchronous frontal theta waves [37], which closely relates to techniques for brain activity description in this study.

*Limitations of the Study*

One of the limitations of this research is that a BCI is required. The ASD user should not have much hair. The BCI must be pleasant and tolerated by them. Moreover, the electrodes must be kept hydrated with saline solution. It also depends on the battery life of the BCI. The emotional state of the ASD user is essential because good measurements will not be obtained if altered. Activities should be done in a scenario with learning conditions without distractions, such as a classroom.

## 5. Conclusions

In this paper, a methodology for the classification of attention by EEG signals of an ASD user was presented. The EEG data acquisition was performed while the ASD user performed some didactic learning activities. In addition, our dataset was created for the post-processing of the information and training of the ML algorithms. To create the dataset, it was necessary to perform preprocessing, filtering, and feature extraction. The proposed features can be used to train and evaluate several ML models to classify attention using EEG signals.

On the other hand, with these findings, therapists, teachers, and psychologists can develop better learning scenarios according to the cognitive needs of ASD users. In addition, diagnosis accuracy can be improved by acquiring individual EEG features, which provide relevant information for differential clinical neurodevelopmental symptomatology classification. Furthermore, with the proposed methodology, one can obtain quantifiable information about the performance of ML models when an ASD user performs didactic/learning activities, the above with the purpose of reinforcing the perception of the teacher or therapist.

The future work will involve implementing the proposed method on a real-time embedded system—for example, a stand-alone version using an edge device, novel deep learning methods, and internet of things (IoT). It is possible to explore the feasibility of a mobile-based platform that links with a BCI, instead of a computer. Furthermore, future replication of this methodology is needed to approach a broad spectrum of attention processes and standard estimation.

**Supplementary Materials:** The following supporting information can be downloaded at: https://www.mdpi.com/article/10.3390/mca27020021/s1.

**Author Contributions:** Conceptualization, R.J.-R. and E.I.-G.; Data curation, L.J.-B. and A.S.-T.; Formal analysis, E.E.G.-G., L.J.-B. and A.S.-T.; Funding acquisition, J.J.E.-E. and O.R.L.-B.; Investigation, J.J.E.-E. and E.T.-C.; Methodology, J.J.E.-E., G.M.G.-A. and E.I.-G.; Project administration, O.R.L.-B.; Resources, J.J.E.-E. and O.R.L.-B.; Software, J.J.E.-E.; Supervision, R.J.-R. and E.I.-G.; Validation, G.M.G.-A.; Visualization, A.S.-T.; Writing—original draft, J.J.E.-E.; Writing—review and editing, E.E.G.-G., E.T.-C. and E.I.-G. All authors have read and agreed to the published version of the manuscript.

**Funding:** This research had funds provided by the Universidad Autónoma de Baja California (UABC) through the grants number 679 and 300/2610.

**Institutional Review Board Statement:** The study was conducted according to the guidelines of the Declaration of Helsinki, and approved by the Ethics Committee and Research for Pre-Graduates and Post-Graduates of the Facultad de Ingeniería y Negocios Guadalupe Victoria of the Universidad Autónoma de Baja California; it was approved on 8 October 2020, with the POSG/020-1-04 code.

**Informed Consent Statement:** Written informed consent has been obtained from the patient to publish this paper.

**Data Availability Statement:** We share the dataset as Supplementary Material.

**Acknowledgments:** We want to thank Escudero Garrido Elena Patricia (Interdisciplinary Professional Unit of Biotechnology of the National Polytechnic Institute, UPIBI-IPN), Gutiérrez Montiel Ixchel (Interdisciplinary Professional Unit of Biotechnology of the National Polytechnic Institute, UPIBI-IPN), Macarena de Haro Iliana Elizabeth (Interdisciplinary Professional Unit of Biotechnology of the National Polytechnic Institute, UPIBI-IPN), Martínez Amaya Laura Fernanda (Manizalez Autonomous University, Colombia), Martínez Oliva Gonzalo Guillermo (Interdisciplinary Pro-fessional Unit of Biotechnology of the National Polytechnic Institute, UPIBI-IPN), López Rivas Andrea (Autonomous University of Baja California, UABC) and Becerril Valenzuela Brando (Autonomous University of Baja California, UABC) and Martinez Verdin Annette Sofia (Autonomous University of Baja California, UABC), for their participation in this project via the Summer of Scientific and Technological Research of the Pacific or Research Activities of UABC. We want to thank the Chemical Sciences and Engineering Faculty of the Autonomous University of Baja California (UABC) for supporting the project with grant number 300/2610. The authors would like to thank INAOE for accepting researcher Everardo Inzunza-González to carry out his sabbatical stay. Thanks are given to PRODEP (Professional Development Program for Professors) for supporting the academic groups to increase their degree of consolidation. We also want to thank the Tijuana Special Education School Eduke and the teachers Jessica Avelar, Letycia Gutiérrez, and Atziri Torres, for supplying the activity sheets and guiding us in working with the ASD person.

**Conflicts of Interest:** The authors declare no conflict of interest. The funders had no role in the design of the study; in the collection, analyses, or interpretation of data; in the writing of the manuscript, or in the decision to publish the results.

## Abbreviations

The following abbreviations are used in this manuscript:

| | |
|---|---|
| AI | Artificial Intelligence |
| ANFIS | Adaptive Neuro-Fuzzy Inference System |
| ASD | Autism Spectrum Disorder |
| BS | Breeding Swarm |
| CNN | Convolutional Neural Networks |
| DL | Deep Learning |
| fMRI | Functional Magnetic Resonance Imaging |
| GAN | Generative Adversarial Networks |
| LSTM | Long Short-Term Memories |
| MBP | Mindfulness-Based Program |
| ML | Machine Learning |
| MRI | Magnetic Resonance Imaging |
| PSD | Power Spectrum Density |
| RAP | Relative Alpha Power |
| RBP | Relative Beta Power |
| RNN | Recurrent Neural Network |
| RTP | Theta Relative Power |
| TAR | Theta–Alpha Ratio |
| TBR | Theta–Beta Ratio |
| TD | Typically Developed |

## Appendix A. Comparison of the Proposed Method with the State of the Art

**Table A1.** State of the art, part 1.

| Reference | Dataset | Data Source | Preprocessing | Method/Algorithm | Main Findings | Application |
|---|---|---|---|---|---|---|
| Ref. [20] | 12 ASD users and 12 typical children | EEG | N/A | Trial-averaged phase-locking value (PLV) approach and cubic support vector machine (SVM) | 95.8% Accuracy, 100% Sensitivity, and 92% Specificity | ASD Classification/Detection |
| Ref. [27] | 97 children aged from 3 to 6 | EEG, eye-tracking tests individually on own-race and other-race stranger faces stimuli. | Data were band-pass filtered between 0.5 and 45 Hz. To improve computing speed, EEG data were then down-sampled to 250 Hz. Power line noise in EEG was removed by a notch filter centered at 50 Hz. Artifacts in EEG were removed using an ICA approach (EEGLab). | SVM, minimum-redundancy-maximum-relevance (MRMR). | Classification Accuracy from combining two types of data reached a maximum of 85.44%, AUC 0.93, when 32 features were selected. | ASD Classification/Detection |
| Ref. [28] | 10 typically developing children (6 Male and 6 Female) and 10 autistic children (6 Male and 4 Female). | Natus Nihon Ohden MEB9000 version 05–81. | 22 channels, sampling frequency of 500 Hz and filtered with a low pass filter and a high pass filter at a frequency range of [0.53, 70] Hz. After filtering out the signal at a frequency range of 0.53 to 70 Hz, the ocular artifacts in the EEG signal were removed by thresholding. The threshold was set based on the average value of the amplitude of the eye blink signal. The eye blink signal was observed for 10 seconds with the eye open and eye close event. | ResNet50 | Average Accuracy of 81% | ASD Classification/Detection |
| Ref. [29] | N/A | EEG | Wavelet Transform, reduction of dimensionality, removal of irrelevant data. | K-Nearest Neighbour (KNN), A Correlation-based Feature Selection (CFS), Minimum Redundancy Maximum Relevance (MRMR) and the Information Gain (IG). | N/A | ASD Classification/Detection |
| Ref. [30] | Dataset from King Abdulaziz University (KAU) Hospital, Saudi Arabia. It is a public available dataset found in Sixteen subjects with twelve from ASD group (3 girls and 9 boys, age 6–20 years old) and four subjects from control group (all boys, 9–13 years old). | Spectrogram of EEG | Artifacts were removed from raw EEG data with re-referencing, filtering and normalization. Common average referencing (CAR) is used for re-referencing. IIR filter is used to low pass filter the signal at 40 Hz cut off frequency and finally the filtered signals from each electrode is normalized to the interval $[-1, 1]$. signals are segmented into 3.5 second window frames for each subject to the dataset. Using Short-Time Fourier Transform (STFT) for each of the above segments, the spectrogram plot is generated in the last step and saved as image. | NB, LDA, RF, kNN, LR and SVM. Ten-fold cross validation. Three different CNN models. | The proposed DL based model achieves higher accuracy (99.15%) compared to the ML based model (95.25%) on an ASD EEG dataset and also outperforms existing methods. | ASD Classification/Detection |
| Ref. [31] | 5 Neurotypical and 8 ASD. | EEG | High-pass filtered at 1 Hz to remove slow trends and subsequently low-pass filtered at 50 Hz to remove line noise. The routine clinical bandwidth for EEG is from 0.5 to 50 Hz. | ML classifiers, namely support vector machine (SVM) and deep learning methods. | Multiclass two-layer LSTM RNN deep learning classifier is capable of identifying mental stress from ongoing EEG with an overall accuracy of 93.27%. | ASD Mental Stress |
| Ref. [32] | Study 1, 15 teenagers. Study 2, 20 subjects diagnosed with ASD and 20 subjects diagnosed with other neuropsychiatric disorders. | Artifact-free EEG data. | Raw EEG time-series were analyzed with a features extraction algorithm to extract 794 quantitative features (TSFRESH Python package). | TWIST, Sine-net ANN and Back Propagation ANN. | Sine-net ANN reached the best predictive capability in distinguishing autistic cases from typicals in study 1, Accuracy of 100%. Back Propagation ANN reached the best predictive capability in distinguishing autistic cases from subjects affected by other neuropsychiatric disorders in study 2 with an overall accuracy of 94.95%. | ASD Classification/Detection |

**Table A2.** State of the art, part 2.

| Reference | Dataset | Data Source | Preprocessing | Method/Algorithm | Main Findings | Application |
|---|---|---|---|---|---|---|
| Ref. [33] | 122 subjects. | EEG to image converted | Spectrogram image generation model is presented using a combination of 1D_LBP and STFT. | Decision Tree (DT), Discriminant Analysis (DA), Logistic Regression (LR), SVM, K-Nearest Neighbor (kNN). | SVM classifier reached 96.44% Accuracy | ASD Classification/Detection |
| Ref. [38] | The disorder group comprises of 8 boys (10–16 years), and the normal group consists of 10 boys (9–16 years). Neuroimaging: ABIDE-I and ABIDE-II, which encompasses sMRI, rs-fMRI, and phenotypic data. ABIDE-I: a total of 1112 datasets, 539 individuals with ASD and 573 healthy individuals (ages 64–7). ABIDEII: 1114 datasets from 521 individuals with ASD and 593 healthy individuals (ages 5–64). | Neuroimaging, fMRI, MRI, EEG | Several preprocesing applied to all signals. | Supervised learning, unsupervised learning, and reinforcement learning (RL). | Presents challenges and performances of DL techniques | ASD Classification/ Rehabilitation |
| Ref. [39] | MICCAI 2008, MICCAI 2016, ISBI 2015, and eHealth Lab. | MRI | Low level and high level preprocessing methods in MRI. | Most popular DL architectures for MS detection: convolutional neural networks (CNNs), Autoencoders (AEs), generative adversarial networks (GANs), and CNN-RNN models. | The inaccessibility of huge sMRI datasets belonging to a diverse population and lack of access to fMRI modalities are among the most important dataset-related challenges which are discussed in detail. Moreover, DL-related challenges include researchers' lack of access to powerful hardware resources for MS diagnosis research. | Multiple Sclerosis Detection |
| Ref. [40] | Dataset of the Institute of Psychiatry and Neurology in Warsaw. | EEG | EEG signals were divided into 25 s time frames and then normalized by z-score or norm L2. | EEG signals Classification: support vector machine, k-nearest neighbors, Decision Tree, Naive Bayes, Random Forest, extremely randomized trees, and bagging. DL models: long short-term memories (LSTMs), one-dimensional convolutional networks (1D-CNNs), and 1D-CNN-LSTM. | CNN-LSTM model accuracy of 99.25%. | Schizophrenia/Diagnosis |
| Ref. [41] | Bonn University dataset with six classification combinations and the Freiburg dataset. | EEG | Tunable-Q wavelet transform (TQWT) for EEG signal decomposition. Feature extraction, 13 different fuzzy entropies calculated from TQWT. Six layers Autoencoder (AE) for dimensionality reduction. | Classification: Adaptive neuro-fuzzy inference system (ANFIS), and also its variants with grasshopper optimization algorithm (ANFIS-GOA), particle swarm optimization (ANFIS-PSO), and breeding swarm optimization (ANFIS-BS). | ANFIS-BS two classes classification Accuracy: 99.74%; an Accuracy of 99.46% in ternary classification on the Bonn dataset, and 99.28% on the Freiburg dataset. | Detection of epileptic seizures |
| Proposed method | 1 Subject, 33936 samples | EEG and BCI | Scaling, 2 seconds Band Power Separation. Features: TRP, ARP, BRP, TBR, TAR, and Theta/(alpha+beta). | Naive Bayes, Stochastic Gradient Descent, Decision trees, SVM, KNN, MLP-NN, RF, Extra trees. | (MLP-NN) with Accuracy 92.98%, Sensitivity 94.59%, F1 Score 93.04%, Specificity 91.55%, AUC of 0.9299, Cohen's Kappa coefficient of 0.8597, Matthews correlation coefficient of 0.8602, and Hamming loss of 0.0701. | ASD Attention Classification |

## Appendix B. Fragment of the Dataset Created for This Study

| | A | B | C | D | E | F | G | H | I | J | K | L | M | N | O | P | Q | R | S | T | U | V | W | X | Y |
|---|---|---|---|---|---|---|---|---|---|---|---|---|---|---|---|---|---|---|---|---|---|---|---|---|---|
| 1 | Rtp_F3 | Rap_F3 | Rbp_F3 | Tbr_F3 | Tar_F3 | Tbar_F3 | Rtp_P7 | Rap_P7 | Rbp_P7 | Tbr_P7 | Tar_P7 | Tbar_P7 | Rtp_F4 | Rap_F4 | Rbp_F4 | Tbr_F4 | Tar_F4 | Tbar_F4 | Rtp_P8 | Rap_P8 | Rbp_P8 | Tbr_P8 | Tar_P8 | Tbar_P8 | Classes |
| 2 | 0.0102257 | 0.0080753 | 0.9816991 | 0.0104163 | 0.0082258 | 0.0103313 | 0.0182301 | 0.0173039 | 0.964466 | 0.0189017 | 0.0179414 | 0.0185686 | 0.4418397 | 0.3412736 | 0.2168867 | 2.0371914 | 1.5735111 | 0.7916 | 0.3602113 | 0.1162547 | 0.523534 | 0.6880381 | 0.2220576 | 0.5630161 | 0 |
| 3 | 0.0124157 | 0.0081978 | 0.9793865 | 0.012677 | 0.0083704 | 0.0125717 | 0.019533 | 0.0170307 | 0.9634362 | 0.0202743 | 0.0176771 | 0.0199222 | 0.4339794 | 0.35182 | 0.2142006 | 2.0260418 | 1.6424793 | 0.76672 | 0.357329 | 0.1170774 | 0.5255935 | 0.6798581 | 0.2227528 | 0.5560062 | 0 |
| 4 | 0.0137813 | 0.0079854 | 0.9782333 | 0.0140879 | 0.0081631 | 0.0139739 | 0.0228004 | 0.0164545 | 0.9607451 | 0.023732 | 0.0171268 | 0.0233324 | 0.4158206 | 0.3685092 | 0.2156702 | 1.9280392 | 1.7086701 | 0.7118029 | 0.3554891 | 0.1135679 | 0.530943 | 0.6695429 | 0.2138984 | 0.5515642 | 0 |
| 5 | 0.0137383 | 0.0075644 | 0.9786973 | 0.0140373 | 0.007729 | 0.0139297 | 0.0279555 | 0.0159336 | 0.9561109 | 0.0292388 | 0.016665 | 0.0287595 | 0.4025978 | 0.3809536 | 0.2164486 | 1.8600157 | 1.7600186 | 0.6739142 | 0.3551171 | 0.1080237 | 0.5368592 | 0.6614716 | 0.2012143 | 0.5506691 | 0 |
| 6 | 0.0127689 | 0.007206 | 0.9800251 | 0.0130292 | 0.0073528 | 0.0129341 | 0.0348482 | 0.0166259 | 0.9485259 | 0.0367393 | 0.0175282 | 0.0361064 | 0.4040353 | 0.3820412 | 0.2139235 | 1.8886906 | 1.7858775 | 0.6779518 | 0.343822 | 0.0985567 | 0.5576213 | 0.6165869 | 0.1767449 | 0.5239767 | 0 |
| 7 | 0.0114414 | 0.0068034 | 0.9817553 | 0.011654 | 0.0069298 | 0.0115738 | 0.0422853 | 0.0195124 | 0.9382023 | 0.0450706 | 0.0207976 | 0.0441523 | 0.431724 | 0.3619059 | 0.20637 | 2.0919899 | 1.7536748 | 0.7597084 | 0.3205963 | 0.0889285 | 0.5904752 | 0.5429463 | 0.150605 | 0.4718789 | 0 |
| 8 | 0.0102782 | 0.0061561 | 0.9835657 | 0.01045 | 0.006259 | 0.010385 | 0.0499361 | 0.0257875 | 0.9242764 | 0.0540272 | 0.0279003 | 0.0525608 | 0.4715165 | 0.3270261 | 0.2014574 | 2.3405267 | 1.6233016 | 0.8922065 | 0.28744 | 0.0850815 | 0.6274785 | 0.4580875 | 0.1355927 | 0.4033907 | 0 |
| 9 | 0.0096363 | 0.0053189 | 0.9850448 | 0.0097826 | 0.0053997 | 0.00973 | 0.0564655 | 0.0347005 | 0.908834 | 0.0621296 | 0.0381814 | 0.0598447 | 0.5021135 | 0.2932252 | 0.2046613 | 2.453388 | 1.4327344 | 1.0084899 | 0.257834 | 0.0903469 | 0.6518191 | 0.3955606 | 0.1386073 | 0.3474074 | 0 |
| 10 | 0.0097321 | 0.0045489 | 0.985719 | 0.0098731 | 0.0046148 | 0.0098278 | 0.060201 | 0.0437183 | 0.8960807 | 0.0671826 | 0.0487884 | 0.0640574 | 0.5034565 | 0.2790178 | 0.2175257 | 2.3144689 | 1.2826886 | 1.0139223 | 0.2474764 | 0.1038717 | 0.6486519 | 0.3815243 | 0.1601347 | 0.328862 | 0 |
| 11 | 0.0102577 | 0.0039654 | 0.9857768 | 0.0104057 | 0.0040226 | 0.010364 | 0.0593589 | 0.0494421 | 0.891199 | 0.0666057 | 0.0554782 | 0.0631047 | 0.4696732 | 0.2949334 | 0.2353934 | 1.9952689 | 1.2529384 | 0.8856296 | 0.2665594 | 0.1194774 | 0.6139631 | 0.4341619 | 0.1946004 | 0.363437 | 0 |
| 12 | 0.0108793 | 0.0035984 | 0.9855223 | 0.0110391 | 0.0036513 | 0.0109989 | 0.0546509 | 0.0507077 | 0.8946414 | 0.0610869 | 0.0566794 | 0.0578102 | 0.4092739 | 0.3387292 | 0.2519969 | 1.6241226 | 1.3441801 | 0.6928318 | 0.3125059 | 0.1304666 | 0.5570275 | 0.5610241 | 0.2342193 | 0.4545579 | 0 |
| 13 | 0.0114132 | 0.0034962 | 0.9850905 | 0.011586 | 0.0035492 | 0.011545 | 0.0476615 | 0.0480615 | 0.904277 | 0.0527068 | 0.0531491 | 0.0500468 | 0.3399286 | 0.3990302 | 0.2610412 | 1.3022029 | 1.5286102 | 0.5149876 | 0.3724082 | 0.1328731 | 0.4947188 | 0.7527674 | 0.268583 | 0.5933923 | 0 |
| 14 | 0.011693 | 0.0037608 | 0.9845462 | 0.0118766 | 0.0038199 | 0.0118314 | 0.0406044 | 0.0429516 | 0.916444 | 0.0443065 | 0.0468677 | 0.0423229 | 0.2864039 | 0.4581543 | 0.2554418 | 1.1212101 | 1.7935761 | 0.401353 | 0.4293552 | 0.1252955 | 0.4453492 | 0.9640866 | 0.2813422 | 0.7524037 | 0 |
| 15 | 0.0112239 | 0.0044795 | 0.9842966 | 0.011403 | 0.004551 | 0.0113513 | 0.0354243 | 0.0368054 | 0.9277703 | 0.0381821 | 0.0396708 | 0.0367252 | 0.2664664 | 0.4992771 | 0.2342566 | 1.1374979 | 2.1313259 | 0.363264 | 0.465237 | 0.1096699 | 0.425093 | 1.0944358 | 0.2579905 | 0.8699874 | 0 |
| 16 | 0.0094187 | 0.0053526 | 0.9852287 | 0.0095599 | 0.0054328 | 0.0095083 | 0.0330734 | 0.030314 | 0.9366126 | 0.0353117 | 0.0323655 | 0.0342047 | 0.300691 | 0.5009614 | 0.1983476 | 1.5159799 | 2.5256743 | 0.429983 | 0.461863 | 0.0911257 | 0.4470112 | 1.0332247 | 0.2038556 | 0.858263 | 0 |
| 17 | 0.0070177 | 0.0056709 | 0.9873114 | 0.0071079 | 0.0057438 | 0.0070673 | 0.034861 | 0.0249049 | 0.9402342 | 0.0370769 | 0.0264879 | 0.0361201 | 0.3601675 | 0.4763241 | 0.1635084 | 2.2027454 | 2.9131465 | 0.562909 | 0.4272422 | 0.0782682 | 0.4944896 | 0.8640063 | 0.1582808 | 0.7459386 | 0 |
| 18 | 0.005506 | 0.0053482 | 0.9891459 | 0.0055664 | 0.0054068 | 0.0055365 | 0.038034 | 0.0197272 | 0.9422388 | 0.0403656 | 0.0209365 | 0.0395378 | 0.4430177 | 0.4226684 | 0.1343139 | 3.2983748 | 3.1468695 | 0.7953891 | 0.3617783 | 0.072336 | 0.5658858 | 0.6393132 | 0.1278278 | 0.5668536 | 0 |
| 19 | 0.0049982 | 0.0047888 | 0.990213 | 0.0050476 | 0.0048361 | 0.0050233 | 0.0435218 | 0.0171621 | 0.9393161 | 0.0463335 | 0.0182709 | 0.0455021 | 0.5006572 | 0.3778448 | 0.1214979 | 4.1207059 | 3.1098871 | 1.0026324 | 0.3062104 | 0.0743903 | 0.6193993 | 0.4943667 | 0.1201007 | 0.4413592 | 0 |
| 20 | 0.0051411 | 0.0043519 | 0.990507 | 0.0051904 | 0.0043936 | 0.0051677 | 0.0460258 | 0.016129 | 0.9378453 | 0.0490761 | 0.0171979 | 0.0482463 | 0.5447706 | 0.3380112 | 0.1172182 | 4.6474925 | 2.8836074 | 1.1966947 | 0.2644763 | 0.0772228 | 0.6583008 | 0.401756 | 0.1173063 | 0.3595755 | 0 |
| 21 | 0.0058207 | 0.0041266 | 0.9900527 | 0.0058792 | 0.0041681 | 0.0058548 | 0.0495438 | 0.018041 | 0.9324152 | 0.0531349 | 0.0193487 | 0.0521263 | 0.557043 | 0.320445 | 0.122512 | 4.5468425 | 2.6156203 | 1.2575553 | 0.2509923 | 0.0817746 | 0.667233 | 0.3761689 | 0.1225578 | 0.3350998 | 0 |
| 22 | 0.0073671 | 0.0042264 | 0.9884065 | 0.0074535 | 0.004276 | 0.0074217 | 0.0510271 | 0.0207348 | 0.9282381 | 0.054972 | 0.0223378 | 0.0537709 | 0.564282 | 0.306124 | 0.1295939 | 4.3542324 | 2.3621791 | 1.2950626 | 0.2460561 | 0.0853953 | 0.6685487 | 0.3680451 | 0.1277323 | 0.3263586 | 0 |
| 23 | 0.0101763 | 0.0047204 | 0.9851033 | 0.0103302 | 0.0047918 | 0.0102809 | 0.054881 | 0.0245947 | 0.9205243 | 0.0596193 | 0.0267182 | 0.0580678 | 0.5690571 | 0.2915452 | 0.1393978 | 4.0822544 | 2.0914625 | 1.3204929 | 0.240471 | 0.0905182 | 0.6690108 | 0.3594427 | 0.1353016 | 0.3166055 | 0 |
| 24 | 0.0143229 | 0.0055827 | 0.9800944 | 0.0146138 | 0.0056961 | 0.014531 | 0.060537 | 0.0281295 | 0.9113335 | 0.0664268 | 0.0308663 | 0.0644378 | 0.5847976 | 0.2664661 | 0.1487363 | 3.9317736 | 1.7915336 | 1.4084637 | 0.2237892 | 0.097089 | 0.6791218 | 0.3295273 | 0.1429626 | 0.2883098 | 0 |
| 25 | 0.0187164 | 0.0065112 | 0.9747724 | 0.0192008 | 0.0066797 | 0.0190734 | 0.0698402 | 0.031085 | 0.8990748 | 0.07768 | 0.0345745 | 0.075084 | 0.6071146 | 0.2343 | 0.1585854 | 3.8283129 | 1.4774371 | 1.5452715 | 0.1986098 | 0.1071318 | 0.6942584 | 0.2860748 | 0.1543112 | 0.2478316 | 0 |
| 26 | 0.0211509 | 0.0069246 | 0.9719245 | 0.0217619 | 0.0071246 | 0.0216079 | 0.0821744 | 0.0329856 | 0.88484 | 0.0928692 | 0.0372786 | 0.0895316 | 0.6330781 | 0.1986249 | 0.168297 | 3.7616724 | 1.1802047 | 1.7253758 | 0.172614 | 0.1231824 | 0.7042036 | 0.2451195 | 0.1749244 | 0.2086257 | 0 |
| 27 | 0.0209215 | 0.0065909 | 0.9724876 | 0.0215134 | 0.0067774 | 0.0213686 | 0.0943953 | 0.0338323 | 0.8717724 | 0.1082798 | 0.0388087 | 0.1042346 | 0.6522901 | 0.1683934 | 0.1793165 | 3.6376471 | 0.9390847 | 1.8759609 | 0.1493887 | 0.1447674 | 0.705844 | 0.2116455 | 0.2050982 | 0.1756251 | 0 |
| 28 | 0.0199334 | 0.0059587 | 0.9741079 | 0.0204633 | 0.006117 | 0.0203388 | 0.1072521 | 0.0361022 | 0.8566458 | 0.1252 | 0.0421436 | 0.120137 | 0.6477593 | 0.1523461 | 0.1998946 | 3.2405039 | 0.7621322 | 1.8389675 | 0.1334291 | 0.1740861 | 0.6924848 | 0.1926816 | 0.2513934 | 0.1539736 | 0 |
| 29 | 0.0202276 | 0.005485 | 0.974873 | 0.0207615 | 0.0056298 | 0.0206453 | 0.1101597 | 0.03885 | 0.8509903 | 0.1294488 | 0.0456527 | 0.1237971 | 0.6242423 | 0.1454705 | 0.2302872 | 2.7107118 | 0.6316916 | 1.6612893 | 0.1252188 | 0.206933 | 0.6678482 | 0.1874959 | 0.3098504 | 0.143143 | 0 |
| 30 | 0.0223704 | 0.0052405 | 0.9723891 | 0.0230056 | 0.0053893 | 0.0228823 | 0.1083944 | 0.0440203 | 0.8475853 | 0.1278861 | 0.0519362 | 0.1215721 | 0.5776638 | 0.1451681 | 0.2771681 | 2.0841638 | 0.5237547 | 1.3677817 | 0.1272812 | 0.2439911 | 0.6287277 | 0.2024424 | 0.3880712 | 0.1458444 | 0 |
| 31 | 0.0260355 | 0.0051935 | 0.968771 | 0.0268748 | 0.0053609 | 0.0267315 | 0.0985104 | 0.0478026 | 0.853687 | 0.1153941 | 0.0559954 | 0.1092751 | 0.542735 | 0.1392828 | 0.3179822 | 1.7068097 | 0.4380209 | 1.1869157 | 0.134648 | 0.2749728 | 0.5903792 | 0.2280705 | 0.4657563 | 0.1555992 | 0 |
| 32 | 0.031128 | 0.0055411 | 0.9633309 | 0.0323129 | 0.005752 | 0.0321281 | 0.0877064 | 0.0506965 | 0.8615971 | 0.1017952 | 0.0588402 | 0.0961384 | 0.5093662 | 0.1363757 | 0.3542581 | 1.4378394 | 0.3849615 | 1.0381801 | 0.1475369 | 0.3010823 | 0.5513808 | 0.2675771 | 0.5460515 | 0.1730713 | 0 |
| 33 | 0.0354484 | 0.0063008 | 0.9582508 | 0.0369928 | 0.0065753 | 0.0367511 | 0.0789929 | 0.0530242 | 0.8679829 | 0.0910074 | 0.061089 | 0.0857679 | 0.4969496 | 0.1333259 | 0.3697245 | 1.344108 | 0.3606089 | 0.9878724 | 0.1611231 | 0.3151137 | 0.5237632 | 0.3076258 | 0.601634 | 0.19207 | 0 |
| 34 | 0.0358521 | 0.0074341 | 0.9567138 | 0.0374742 | 0.0077704 | 0.0371853 | 0.0710942 | 0.0532618 | 0.875644 | 0.0811908 | 0.0608259 | 0.0765354 | 0.4774386 | 0.1361777 | 0.3863838 | 1.2356589 | 0.3524414 | 0.9136506 | 0.1770675 | 0.3219488 | 0.5009837 | 0.3534396 | 0.6426332 | 0.2151665 | 0 |
| 35 | 0.0298397 | 0.0084185 | 0.9617418 | 0.0310268 | 0.0087534 | 0.0307575 | 0.0685476 | 0.0543052 | 0.8771472 | 0.0781483 | 0.0619111 | 0.0735922 | 0.4555648 | 0.1413245 | 0.4031107 | 1.1301232 | 0.3505847 | 0.8367659 | 0.1928804 | 0.3212049 | 0.4859148 | 0.3969428 | 0.6610313 | 0.2389737 | 0 |
| 36 | 0.0210839 | 0.0090167 | 0.9698994 | 0.0217382 | 0.0092966 | 0.021538 | 0.0677271 | 0.0523913 | 0.8798816 | 0.076973 | 0.0595436 | 0.0726473 | 0.4346541 | 0.1465537 | 0.4187922 | 1.0378753 | 0.3499438 | 0.7688285 | 0.2020514 | 0.3086883 | 0.4892603 | 0.4129733 | 0.6309286 | 0.2532136 | 0 |
| 37 | 0.0135885 | 0.0092488 | 0.9771627 | 0.013906 | 0.0094649 | 0.0137757 | 0.0730188 | 0.0494948 | 0.8774864 | 0.0832136 | 0.0564052 | 0.0787705 | 0.4290578 | 0.1468533 | 0.4240889 | 1.0117166 | 0.3462795 | 0.7514908 | 0.1968479 | 0.2823358 | 0.5208163 | 0.3779604 | 0.5421025 | 0.2450942 | 0 |
| 38 | 0.0088801 | 0.0092361 | 0.9818838 | 0.0090439 | 0.0094065 | 0.0089596 | 0.0781066 | 0.0414949 | 0.8803985 | 0.0887173 | 0.047132 | 0.0847241 | 0.4606392 | 0.1368442 | 0.4025166 | 1.1443981 | 0.3399716 | 0.8540465 | 0.174063 | 0.2430136 | 0.5829234 | 0.2986035 | 0.4168878 | 0.2107461 | 0 |
| 39 | 0.0069396 | 0.0091711 | 0.9838893 | 0.0070532 | 0.0093213 | 0.006988 | 0.0780478 | 0.0309431 | 0.8910091 | 0.0875949 | 0.0347282 | 0.084655 | 0.5204878 | 0.1205016 | 0.3590107 | 1.4497836 | 0.335649 | 1.0854526 | 0.1460077 | 0.2065428 | 0.6474495 | 0.2255121 | 0.3190099 | 0.1709708 | 0 |
| 40 | 0.0071394 | 0.0090914 | 0.9837691 | 0.0072572 | 0.0092414 | 0.0071908 | 0.0680201 | 0.0221527 | 0.9098271 | 0.0747616 | 0.0243483 | 0.0729845 | 0.5896496 | 0.103153 | 0.3071975 | 1.919448 | 0.3357872 | 1.4369415 | 0.1242805 | 0.1812023 | 0.6945172 | 0.1789451 | 0.2609041 | 0.1419181 | 0 |
| 41 | 0.0091097 | 0.0090057 | 0.9818846 | 0.0092778 | 0.0091719 | 0.0091935 | 0.0542515 | 0.0183296 | 0.927419 | 0.0584972 | 0.0197641 | 0.0573635 | 0.6461257 | 0.0902613 | 0.263613 | 2.4510391 | 0.3424008 | 1.8258624 | 0.1125915 | 0.1668085 | 0.7206 | 0.1562469 | 0.2314855 | 0.1268768 | 0 |

**Figure A1.** Fragment of the dataset created for this study.

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
