# Peer review of "Attention Measurement of an Autism Spectrum Disorder User Using EEG Signals: A Case Study"

_mca, doi:10.3390/mca27020021_

Round 1

Reviewer 1 Report

In this study, attention was measured for ASD using EEG signals. The main theme of the work is good. That's why I like the idea. But there are some shortcomings. These are below:

  • How many observations does the dataset contain?
  • Since it is a case study, only one subject was taken into account. Can the number of subjects be increased? 
  • A classification method is proposed in the study. But with little data, is this classification satisfactory?

I like this work. Purpose is good. But a little more detail about the data should be shared. 

Reviewer 2 Report

This papers is about Attention Measurement of an Autism Spectrum Disorder User using EEG Signals: A case study. my comments are as follows:
1. The abstract section need to edit. 
2. The introduction section is not well organized and needs to be rewritten. The authors should add the important artificial intelligence methods in the introduction section. In this section, the important methods should be mentioned (advantages and disadvantages), and they should be compared with the proposed method. 
3. The literature review section should be edited. The literature review is not comprehensive. It is better to have a tabular summary of the paper's review to give readers a better understanding of the research done in this field. In this section, some articles should be presented in the form of text, and the rest of the articles should be summarized in the table with this information (Works, Dataset, preprocessing, main methods, Performance (%)). Also, In the Frist paragraph of this section, Please explain more about deep learning methods for medical applications. I recommended some important references as follows:
https://doi.org/10.1016/j.compbiomed.2021.104949
https://doi.org/10.1016/j.bspc.2021.103417
https://doi.org/10.3389/fninf.2021.777977
https://doi.org/10.1016/j.compbiomed.2021.104697 
4. The research question(s) need to appear stronger and clearer.
5. Please clarify your initial hypothesis.
6. Please summarize the "Performance Metrics" in a table. 
7. In discussions you need to critically discuss your work/results against your hypothesis. 
8. Identify the main findings and justify the novelty and contribution of the work.
9. A recap of all the relevant parameters with their meaning should be added to help the reader.
10.  Please highlight the clinical significance of your findings. 
11. Please add a section about "limitation of study". 
12. In the Conclusion section, please explain more about future works. This section requires further discussion.. For example, you can discuss about new deep learning methods.  
13. Please add a table in conclusion and compare your proposed method with another related works. 
14. English language is acceptable in general, but there are some errors that should be corrected.

Round 2

Reviewer 1 Report

Thank you for your answers. Enough for me. My recommendation is to accept.

Reviewer 2 Report

My suggestion is to accept this article.